# Targeted and Untargeted Metabolomic Analyses Reveal Organ Specificity of Specialized Metabolites in the Model Grass *Brachypodium distachyon*

**DOI:** 10.3390/molecules27185956

**Published:** 2022-09-13

**Authors:** Anna Piasecka, Aneta Sawikowska, Nicolas Jedrzejczak-Rey, Mariola Piślewska-Bednarek, Paweł Bednarek

**Affiliations:** 1Institute of Bioorganic Chemistry, Polish Academy of Sciences, Noskowskiego 12/14, 61-704 Poznań, Poland; 2Institute of Plant Genetics, Polish Academy of Sciences, Strzeszyńska 34, 60-479 Poznań, Poland; 3Department of Mathematical and Statistical Methods, Poznań University of Life Sciences, Wojska Polskiego 28, 60-637 Poznań, Poland

**Keywords:** *Brachypodium distachyon*, metabolomics, specialized metabolites, phenylpropanoids, flavonoids, mass spectrometry

## Abstract

*Brachypodium distachyon*, because of its fully sequenced genome, is frequently used as a model grass species. However, its metabolome, which constitutes an indispensable element of complex biological systems, remains poorly characterized. In this study, we conducted comprehensive, liquid chromatography-mass spectrometry (LC-MS)-based metabolomic examination of roots, leaves and spikes of Brachypodium Bd21 and Bd3-1 lines. Our pathway enrichment analysis emphasised the accumulation of specialized metabolites representing the flavonoid biosynthetic pathway in parallel with processes related to nucleotide, sugar and amino acid metabolism. Similarities in metabolite profiles between both lines were relatively high in roots and leaves while spikes showed higher metabolic variance within both accessions. In roots, differences between Bd21 and Bd3-1 lines were manifested primarily in diterpenoid metabolism, while differences within spikes and leaves concerned nucleotide metabolism and nitrogen management. Additionally, sulphate-containing metabolites differentiated Bd21 and Bd3-1 lines in spikes. Structural analysis based on MS fragmentation spectra enabled identification of 93 specialized metabolites. Among them phenylpropanoids and flavonoids derivatives were mainly determined. As compared with closely related barley and wheat species, metabolic profile of Brachypodium is characterized with presence of threonate derivatives of hydroxycinnamic acids.

## 1. Introduction

Purple false brome (*Brachypodium distachyon* (L.) P. Beauv.; hereafter Brachypodium) is closely related to wheat and barley, making it potentially useful for functional genomics of these crops. Its main advantage as a model plant is the smallest genome found in the Poaceae family comprising five chromosomes spanning over 272 Mbp, in which about 25,000 protein-coding sequences are predicted [1]. In addition, Brachypodium is self-fertile and has a rapid life cycle of 8–10 weeks, depending on the environmental growth conditions [2]. A breakthrough point in Brachypodium research was the genome sequencing of accession Bd21 [3], which contributed to several genetic and genomic resources including Phytozome [4] and Gramene [5] and gave rise to initiatives like BrachyPan (Brachypodium pan-genome) [6]. Consequently, Brachypodium became an object of intense research in many fields serving in understanding interaction of grasses with viruses [7], bacteria [8], fungi [9] and invertebrates [10] as well as their responses to abiotic stresses [11].

Studies in Brachypodium included also metabolomic analyses that have been performed in different biological and physiological contexts. During these studies widely targeted metabolomic analysis has been used to compare the metabolomes of seeds and leaves of Bd21 and Bd3-1 accessions [12] and analysis of respective recombinant inbreed lines enabled identification of quantitative trait loci linked with variation of selected metabolites present in seeds [13]. It has also been shown that metabolomic data correlates with phenotypic variability within different *Brachypodium* species [14]. Correlation of metabolomic with proteomic or transcriptomic data enabled the comprehensive description of Brachypodium reaction to fungal infection [15] and drought [16,17]. Results of metabolomic analysis of Brachypodium in correlation with data on biomass production during drought served in building models for phenotype prediction [18]. Finally, differences in metabolomic response to drought between accessions inhibiting different ecological niches have been described [19]. However, despite these individual reports, the Brachypodium metabolome remains virtually unknown. This particularly concerns specialized metabolites, which in plants are involved in responses to environmental cues, including biotic and abiotic stressors. These compounds may play a role in signalling pathways, regulation of many bioprocesses, or directly deterring antibiotic agents [20]. Concerning these multifarious functions, studies of specialized metabolites are important for investigating the interactions of plants with the environment. Plants collectively produce a large and diverse array of these compounds [21]. Some groups of specialized metabolites have a very restricted distribution, i.e., they are often only found in taxonomically related genera or species. On the other hand, some classes of specialized metabolites, for example phenylpropanoids and flavonoids are conserved among plants. However, even in such cases particular end products of these pathways can be also limited to narrow sets of plant species [22]. Due to this high diversity of plant’s specialized metabolites and their limited occurrence, only a small portion of these compounds is known and covered in available metabolomics databases, which in turn significantly hampers analysis of plant metabolomes.

In this study, we used mass spectrometry (MS) techniques to shed more light on specialized metabolism of Brachypodium accessions Bd21 and Bd3-1. Both these lines originate from Iraq but reveal some differences in their morphology and development [23,24]. Particularly, Bd21 and Bd3-1 strongly differ in their root morphology and changes in root growth in response to low nitrogen and phosphorus supplies [25]. As indicated by earlier studies these two lines clearly differ in their resistance towards viruses [26,27] and fungi [28,29,30,31]. Bd3-1 was more resistant to *Barley stripe mosaic virus* as well as to *Rhizoctonia solani* and *Puccinia emaculata* fungi, while Bd21 appeared to be more resistant to *Ramularia collo-cygni*. Both lines have also different drought tolerance; Bd3-1 is better adopted than Bd21 to cope with this abiotic stress [17,32]. Unlike the earlier metabolomic studies in leaves and seeds of these accessions [12,13], we emphasised compound identification and extended the metabolite analysis to spikes and roots. Particularly this latter underground organ has been shown to significantly differ in specialized metabolite composition from the aerial parts in other plant species [33,34]. This could be also of particular interest regarding the differences between Bd21 and Bd3-1 root morphology [25], which suggests differences between the root metabolite profiles of these lines. Our unbiased metabolic approaches combined with pathway enrichment analysis revealed metabolic pathways that significantly differentiate analysed organs and accessions. In addition, detailed inspection of mass spectra obtained during MS/MS and MS^n^ analyses of Brachypodium extracts combined with database and literature searches enabled preliminary identification of 93 specialized metabolites, mainly phenylpropanoids, produced by this model grass plant.

## 2. Results and Discussion

### 2.1. Comparison of Metabolomics Profiles in Analyzed Brachypodium Organs and Lines

Metabolic diversity within the studied Brachypodium lines and organs was represented in our LC/MS data sets by 22,307 individual signals detected in 48 analysed samples (three organs, two lines, two experiments and four biological replicates). To have a better insight into the global metabolite profile in analysed organs of both Brachypodium lines we performed principal component analysis (PCA) with all *m*/*z* signals detected during analyses performed with high resolution MS system in positive and negative ionization mode. The obtained PC3 plot revealed clear metabolic discrimination among tested organs and relative similarity between both lines (Figure 1). The highest consistency of metabolic profiles was observed within roots of both lines whereas the biggest interline differences were visible for spikes.

To corroborate our observations from the PCA plot, we used univariate two-way ANOVA analysis for each signal to classify signals into three following groups: (i) signals differentiating organs (O: comparison of the mean values of signal intensities from roots, spikes and leaves), (ii) signals differentiating lines (L: comparison of the mean values of signal intensities from Bd21 and Bd3-1) and (iii) signals revealing significant interaction between organ and line factors (L×O: comparison of the mean values of signal intensities from Bd21 roots, spikes and leaves, and Bd3-1 roots, spikes and leaves) (Figure 2A). As already indicated by the PCA plot (Figure 1), there was a relatively low number of signals discriminating lines whereas the majority of signals were organ specific. Nevertheless, the PCA plot was created on the basis of all detected signals while only fraction of them was filtered for O effect after ANOVA. This indicated a high impact of the differentiating signals on the entire metabolomic profiles in Brachypodium plants. Overall, these results are convergent with previous unbiased metabolom analyses conducted on the leaves and seeds of the Bd21 and Bd3-1 lines, which also revealed a stronger impact on organs than the genotype on metabolome [12].

Differences in the metabolite set might contribute to the phenotypic differences between both studied lines, which have proven variation in many phenotypic traits [12,23,25], resistance to particular pathogens [26,27,28,29] or drought adaptation [17,32]. To obtain a better insight into the metabolic differences between Bd21 and Bd3-1 lines we selected signals corresponding to differentially accumulating metabolites (DAMs). We defined DAMs as signals significantly distinguishing both accessions (*p*-value < 0.01 for factors L or L × O) and differing at least two times (fold change; FC > 2) in their abundance in any of the tested organs of Bd3-1 and Bd21 accessions (Figure 2B). Out of all signals, 2295 met these conditions for each organ suggesting a prevalent role for widely occurring elements in line differentiation. As suggested by the PCA plot, the proportion of DAMs indicated the lowest differences between Brachypodium lines in the roots and highest in the spikes. We selected 30 DAMs with the highest diversification among the studied groups to annotate respective *m*/*z* values and compare in detail differences in their abundances between particular lines and organs (Figure 3). Despite good genetic characterization of Brachypodium, the metabolic pathways of this species are fragmentary in all dedicated metabolic platforms. Therefore, annotation of *m*/*z* values was performed with a database created with *Oryza sativa* subsp. *japonica* (japonica rice), described at the metabolome level model plant from Poaceae family [35].

Among the annotated compounds putative derivatives of hydroxycinnamic acids (feruloylhydroxycitric acid, caffeoylpyruvylhexose, isomers of caffeoylthreonic acid and cinnamic acid ethyl ester) were highly represented (Figure 3). These included conjugates of hydroxycitrate with hydroxycinnamic acids known from *Zea mays* as compounds with high variation in accumulation profile in different inbred lines [36]. The signal corresponding to feruloylhydroxycitric acid had the highest abundance in Bd3-1 roots compared with the Bd21 roots. The relatively high level of caffeoylthreonic acid isomers in all organs of the Bd21 line, as compared with Bd3-1, is noteworthy. The same trend of high abundance in Bd21 line was observed for hydroxybenzoic acid derivatives (*N*-salicyloylaspartic acid and *N*-pyruvoyl-methoxy-hydroxyanthranilic acid).

Putative derivatives of the flavone apigenin (isovitexin pentose-deoxyhexoside, pentahydroxy-dimethoxyflavone hexoside and apigenin hydroxy-methylglutaryl-hexoside) together with proanthocyanidin B and cyanidin acylated glycoside were representatives of differentiating flavonoids. Interestingly, cyanidin 3-*O*-glucoside (chrysantemin) has been already reported as differentially accumulating metabolite in Brachypodium spikes [13]. These correlative findings suggest that biosynthesis of cyanidin glycosides clearly discriminate both Brachypodium lines. Despite the common biosynthetic origin, differences in abundance of particular flavonoids and hydroxycinnamic acids were not correlated. However, it should be noted that most of the distinctive compounds were complex structures that were relatively distant from the common precursors in the metabolic pathway. This in turn indicated that the activities of the enzymes responsible for particular modifications of the core structures, including hydroxylation, acylation, methylation and glycosylation, were responsible for the observed differences in phenylpropanoid metabolism among compared organs and lines.

We found signals corresponding to monounsaturated fatty acids (palmitoleic acid, myristoleic acid) with the highest diversification between Bd21 and Bd3-1 lines in leaves. Among annotated phosphate-containing compounds, purine derivatives (deoxyguanosine 5’-monophosphate (dGMP) and adenosine monophosphate) were highly accumulating in spikes of Bd21 in comparison to Bd3-1. Phosphoglycerolipids (PA(20:0/17:1), PS(20:1/0:0), PS(18:1/0:0) and PG(18:0/0:0)) have variable accumulation patterns in all organs and lines, therefore, differences in phosphate management between both lines could be suggested. Finally, sulphur-containing metabolite from the 2-oxocarboxylic acid pathway (methylthio-pentylmalic acid) was also annotated as one of the most differentiated metabolites. Its dominating abundance in Bd3-1 was especially visible in roots.

**Figure 3 molecules-27-05956-f003:**
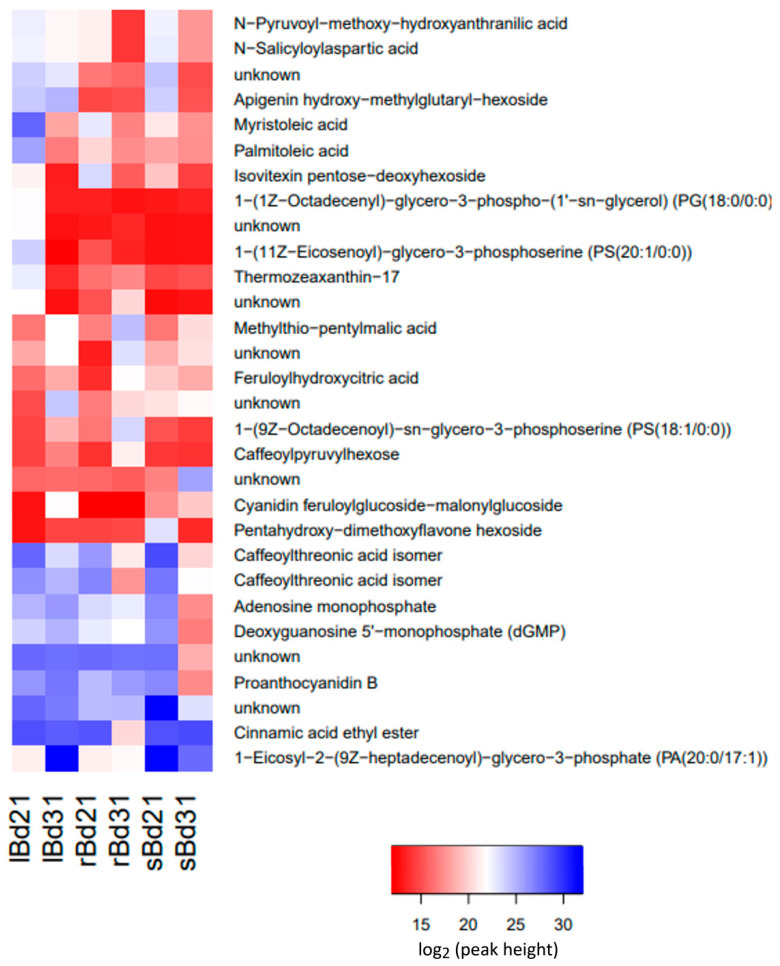
Abundances of peaks representing selected MS signals highly differentiated Brachypodium lines in particular organs. Colour scale presents log_2_ from respective peak heights. Tentative signal annotations based on *O. sativa* metabolic database from Kyoto Encyclopedia of Genes and Genomes (KEGG) [35,37]. l—leaves; r—roots; s—spikes.

### 2.2. Pathway Enrichment Analysis

#### 2.2.1. Most Represented Metabolic Pathways

For further global settling of Brachypodium metabolome into the biological context, functional analysis of obtained result on the basis of MetaboAnalyst was implemented. Firstly, pathway-level enrichment has been performed with all *m*/*z* signals detected in positive and negative ionization modes for overall picture of metabolites in Brachypodium plants (Table 1). The same as for metabolite annotation, our analysis was performed on *O. sativa* database [35]. Direct tentative annotation of *m*/*z* values obtained during our analysis to rice metabolites enabled further calculation of pathway-level enrichment.

Our analysis performed with all signals indicated significant enrichment of flavonoid-related pathways (Table 1). Forty signals have been matched to metabolites from flavonoids biosynthesis (47 metabolites in total). This was accompanied by signals matching 12 metabolites (all from the same pathway) from flavone and flavonol biosynthesis. This indicated a significant contribution of a specialized metabolism to the overall profile of Brachypodium metabolites.

The remaining identified pathways represented primary metabolic processes, mainly nucleotide, sugar and amino acid metabolism. Twenty-five metabolites from the galactose metabolism matched with signals from our analysis, including galactinol and raffinose, which were previously described as involved cold and drought stress response in Brachypodium [38]. Significant annotation of pentose and glucuronate interconversions was mainly related to metabolites from modules of pectin degradation and glucuronate pathway. Key components of plant metabolism from the pentose phosphate pathway as source of substrates for synthesis of purine nucleotides, followed by purine metabolism, were highly matched. In the second mentioned pathway, annotation focused on modules of inosine monophosphate biosynthesis, adenine ribonucleotide biosynthesis and purine degradation. Pyridoxal and pyridoxine, as well as their phosphorylated derivatives from vitamin B6 metabolism, were also significantly enriched. Antioxidant and defence related metabolites from amino sugar and nucleotide sugar metabolism and ascorbate and aldarate metabolism, including phosphorus containing structures (D-Glucosamine phosphate and their derivatives and UDP-glycosides) were significantly annotated.

Amino acid metabolism branched-chain amino acids, valine, leucine and isoleucine biosynthesis were matched. Related to them 2-Oxocarboxylic acid metabolism was also highly scored in both statistical significance and pathway impact. Compounds annotated to this pathway focused on a branch of the 2-Oxocarboxylic acid chain extension by tricarboxylic acid module, which in Brassicaceae species leads to glucosinolate biosynthesis [39]. However, in Poaceae, glucosinolates are absent, therefore, signals annotated to sulphide compounds such as 2-(5’-methylthio)pentylmalic acid and isomer 3-(5’-methylthio)pentylmalic acid), 2-(6’-methylthio)hexylmalic acid and isomer 3-(6’-methylthio)hexylmalic acid from this branch can be components of different pathways. Interestingly, elements of “2-Oxocarboxylic acid” were previously reported in grasses as factor involved in the response to salinity and drought stress [40,41]. Further inspection of LC-MS and MS/MS spectra showed the presence of such *S*-containing compounds in Brachypodium plants not previously identified, which confirms the validity of enrichment analysis in plant metabolite profiling (Appendix A). Another amino acid related pathway, tyrosine metabolism, was selected based on the annotation of tyrosine and 3,4-dihydroxyphenylalanine, which, in grasses, plays a key role in lignin biosynthesis (Maeda, 2016).

#### 2.2.2. Metabolic Pathways Distinguishing Bd21 and Bd3-1 Lines

In order to identify metabolic pathways discriminating on both analysed lines in particular organs, pathway enrichment analysis was only performed for signals representing DAMs selected based on the above-described ANOVA analysis (*p*-value ≤ 0.01 for factor L or O × L; FC > 2) (Figure 2B, Table 2). Housekeeping and general metabolism-related biological pathways (galactose metabolism, pentose phosphate pathway, valine, leucine and isoleucine biosynthesis) highly varied among the six compared groups. Moreover, specialized metabolism of flavonoid biosynthesis, was related to flavone and flavonol biosynthesis and 2-Oxocarboxylic acid metabolism were commonly differentiated these groups.

Metabolic differences between the roots of Bd21 and Bd3-1 are manifested primarily by “Diterpenoid biosynthesis”. In monocots, diterpenoids are known from large structural diversity and species-specificity. In Brachypodium, no specialized diterpenoids have been identified, however, a few Brachypodium genes are homologous to rice genes related to momilactone phytoalexin production [42]. In our analysis, the main differentiating module of diterpenoid biosynthesis was gibberellin production, which is the key factor in root elongation in monocots [43]. The differences in gibberellin levels in the roots of both Brachypodium lines could be related to observed differences in root morphology of both tested lines [25]. In this context it was also of interest that tetrahydrofolate from differentiating one carbon pool by folate pathway has been reported as key regulators of root development [44].

The caffeine metabolism pathway including purine alkaloids was also significantly different among the roots of both Brachypodium lines. Matched intermediates of this pathway included xantosine, 7-methyluric acid and their derivatives. However, caffeine itself has been not reported in grasses while at least some of the matched metabolites can be linked with purine salvage or degradation [45].

Histidine and vitamin B6 metabolism had a shared effect in leaves and spikes. Besides protein synthesis, histidine is tightly connected to nucleotide metabolism and the pentose phosphate pathway. Within this pathway the most differentiated was a branch of histidine biosynthesis from 1-(5-Phospho-d-ribosyl)-ATP via L-histidine to Imidazole-4-acetate. Vitamin B6 metabolism was mainly matched by metabolites from pyridoxal-P biosynthesis branch (pyridoxine, pyridoxine 5-phosphate, pyridoxamine, pyridoxamine 5-phosphate, pyridoxal 5-phosphate, glyceraldehyde 3-phosphate and 4-phosphooxy-threonine, 2-oxo-3-hydroxy-4-phosphobutanoate), which led to further pentose phosphate pathways. This agreed with a previous study showing compounds of vitamin B6 metabolism and their catabolites differently accumulating between Bd21 and Bd3-1 seeds [13].

A pathway that specifically differed between spikes of both Brachypodium lines was the cysteine and methionine metabolism indicating possible differences in sulphate assimilation. *S*-Adenosyl-L-methionine, a key metabolite from this pathway, is a donor of methyl group in numerous transmethylation reaction influencing physical and chemical properties of lignin polymers, as well as hydroxycinnamic acids synthesis in Brachypodium plants [46].

### 2.3. Metabolite Identification with LC-MS Systems

In the next step of our study, we tried to identify a subset of detected metabolites based on their spectra obtained during the HPLC-ESI-MS^n^ and UPLC-HR-MS/MS analyses. MS^n^ spectra are helpful in the identification of complex metabolites, for example flavonoids glycoconjugates, where they can enable the determination of the place and character of the glycosidic bond. In addition, the order of detachment of individual fragments from complex structures with a simultaneous observation of the intensities of particular product ions enables the differentiation of isomeric and isobaric structures, unlike MS/MS, which, in many cases hampers, isomers differentiation. However, accurate measurement of *m*/*z* values obtained during HR MS/MS analysis allowed confirmation of tentative structures predicted by the MS^n^ analysis. Overall, this analysis enabled us to identify 93 metabolites at levels 1–3 according to the Metabolomic Standards Initiative [47] (Table 3). This manual metabolite identification enabled us to describe the structures specific to Brachypodium, which are absent in metabolomic databases and, therefore, cannot be annotated with automated bioinformatics approaches.

#### 2.3.1. Hydroxycinnamoyl-Quinic Acids

MS signals corresponding to phenolic and hydroxycinnamic acids and their derivatives were detected in all studied organs at high abundances. *p*-coumaric, caffeic and ferulic acids have been identified in Brachypodium in conjugation with (methyl)quinic acids, sugar acids, or polyamines; while caffeic and ferulic acids (Table 3; compounds **21**, **38**) have been additionally observed as free molecules that have been identified by comparison to their standards.

Different isomers of hydroxycinnamoyl-quinic acids (HQA) were reported and can be found in metabolite databases, but proper annotation of these isomeric structures should be supported by fragmentation schemes in MS/MS or MS^n^ spectra. In our analysis, HQAs (**8, 26, 29, 43** and **48**) were identified according to Clifford et al. [43] and Piasecka et al. [54]. The main product ion from deprotonated molecules of compounds **29**, **43** and **48** at *m*/*z* = 173 Da corresponded to quinic acid, which is distinctive for 4HQA isomers (Figure 4A). On the other hand, 5HQAs are characterized by the main product ion representing the respective hydroxycinnamic acid molecule [56] as we found for **26** where product ion at *m*/*z* = 193 Da corresponded to ferulic acid (Figure 4B). Deprotonated ions of compounds **8** had the same exact *m*/*z* value as compounds **26** and **43** (367.10425 Da) with the same chemical formula C_17_H_20_O_9_ calculated from exact mass. The UV spectra of **8** with a maximum of absorption at 305 nm (Appendix A) and fragmentation with product ions at *m*/*z* = 161, 135 and 119 Da suggested caffeic acid derivative.

Overall, we observed three different hydroxycinnamic acids conjugated with quinic acid as 4HQA isomers **(29**, **43** and **48**), while only ferulic acid conjugate has been detected as 5HQA isomer (**26**). However, the dominant presence of 4HQAs might result from the isomerization of 5HQAs that could occur either in vivo or in the extract solution [63]. Widely occurring grass 3HQA isomers [56] have been not detected in our study.

#### 2.3.2. Esters of Hydroxycinnamic and Threonic Acids

In addition to quinic acid conjugates, the most abundant signals detected in all studied organs corresponded to sugar acid, mainly threonic acid, derivatives. The main fragmentation pathway of deprotonated ion of compound **42** indicated loss of the 118.0278 Da fragment corresponding to threonic acid moiety [threonic acid-H_2_O]. The main product ion at *m*/*z* = 193.0499 Da revealed loss of the entire ferulic acid moiety (Figure 5A). Less abundant ions were related to the parallel fragmentation scheme in which losses of 176.0489 Da corresponded to [feruloyl-H_2_O]^−^. Two-way fragmentation gave evidence for a similar stability of both acidic components of **42** in CID. Minor product ions at *m*/*z* = 149.0597 and *m*/*z* = 134.0361 Da corresponded to [feruloyl-CO_2_]^−^ and [feruloyl-C_2_O_2_]^−^, indicating a preserved ester bond in fragmentation of **42**. Measurement of the exact mass of ionized compound **42** enabled the calculation of chemical formula C_14_H_15_O_8_, which confirmed the presence of feruloyl and threonate moieties in this structure. Therefore, **42** was determined as feruloylthreonic acid.

The main product ions of compounds **18** and **63** yielded an analogous fragmentation scheme as **42** and the exact mass calculation of product ions indicated caffeic and *p*-coumaric acid residues in those structures, which in turn pointed at caffeoylthreonic and *p*-coumaroylthreonic acids as respective compounds. Analogously to feruloylthreonic acid, expected product ions of deprotonated compound **18** should correspond to [threonic acid-H]^−^ and [caffeic acid-CO_2_]^−^, which were both characterized by the same nominal masses at *m*/*z* = 135 Da. Analysis in the MS^n^ mode with a low resolution of mass measurement could not support discrimination between both product ions. Nevertheless, a high-resolution MS spectra ion at *m*/*z* = 135.0288 (C_4_H_7_O_5_) corresponded to threonate and a second ion at *m*/*z* = 135.04444 Da (C_8_H_7_O_2_) to decarboxylated caffeic acid, confirming the presence of both acid moieties in the structure of compound **18** (Figure 5B). Interestingly, according with our pathway enrichment analysis present in these highly abundant conjugates (**18**, **42**, **63**), threonic acid could be generated as a degradation product of ascorbate in plants [64].

Threonate esters have so far not been reported in Brachypodium or in closely related barley and wheat. However, such compounds were identified in other members of the Poaceae family including silvergrass (*Miscanthus* sp.), orchard grass (*Dactylis glomerata*) and maize [54,65,66]. 2-*O*-caffeoylthreonic acid was previously identified by NMR in *D. glomerata* [65] and *Crataegus* species (Rosaceae) [67]. Moreover, 4-*O*- and 2-*O*-*p*-coumaroylthreonate were observed as precursors in moss cuticle biosynthesis [68]. 2-*O*-caffeoylthreonates with 3-*O*- and 4-*O*-isomers of caffeoylthreonate were also reported in leaves of *Pulmonaria officinalis (Boraginaceae)* [69] and *Fagus*
*Sylvatica* tree (*Fagaceae*) [70], which confirmed structural diversity and the wide presence of these metabolites.

#### 2.3.3. Hydroxycinnamic Acid Amides

Our analysis of MS^n^ and MS/MS spectra also enabled identification of conjugates of *p*-coumaric, caffeic and ferulic acid with putrescine and agmatine. Among these, compounds **10** and **12** have similar masses of protonated [M+H]^+^ ion at *m*/*z* = 235 Da and calculated chemical formula from exact mass as C_13_H_17_O_2_N_2_ (Figure 6A). Compounds **10** and **12** Also share product ion at *m*/*z* = 147 Da that correspond to dehydrated *p*-coumaric acid. In both compounds losses of –NH_4_ group and of entire putrescine moiety can be also observed (Figure 6BC). Fragmentation of compound **12** was characterized by additional product ion at *m*/*z* = 119 Da, which is typical for *p*-coumaric acid with preserved peptide bond between acidic and polyamine substituents.

Regarding this, we tentatively assigned compounds **10** and **12** as *p*-coumaroyl-*N*-putrescine isomeric structures. The same scheme of fragmentation and differences in product ion intensities were observed for compounds **11** and **23**. The protonated [M+H]^+^ ions of both metabolites yielded losses of fragment 88 Da, corresponding to putrescine, whereas the main product ions at *m*/*z* = 248 and 177 Da in an MS2 scan indicated a ferulic acid molecule. In an MS3 scan of compound **23**, an additional product ion at *m*/*z* = 145 Da indicated peptide bond preservation during fragmentation steps. The calculated chemical formula of compounds **11** and **23** and fragmentation spectra corresponded to C_14_H_21_O_3_N_2_, which complied with feruloyl-*N*-putrescine isomers, similar to **10** and **12**. In an MS3 scan of compound **23**, an additional product ion at *m*/*z* = 145 Da indicated peptide bond preservation during fragmentation steps. In grasses, two geometric isomers of hydroxycinnamic acids *cis* and *trans* were previously reported [71]. *Trans* isomers constitute the predominantly occurring form in plants, but they can be transformed to corresponding *cis* isomers by UV-radiation. Concerning this, we assumed that the isomeric hydroxycinnamic acid amides revealing differences in their fragmentation pattern represented *cis-trans* conformers.

Compounds **40** and **49** were observed only in the positive ionization mode. The protonated [M+H]^+^ ions of both molecules yielded losses of the main fragment of 130 Da with product ions at *m*/*z* = 147 and 177 Da, corresponding to *p*-coumaroyl and feruloyl moieties, respectively. The accurate mass measurement suggested C_14_H_19_O_2_N_4_ and C_15_H_21_O_3_N_4_ chemical formulae for **40** and **49**, respectively. Four nitrogen atoms containing a substituent with adequate chemical formulation indicated agmatine presence in those structures. Consequently, compounds **40** and **49** were determined as *p*-coumaroylagmatine and feruloylagmatine. The latter compound has already been reported in Brachypodium leaves [72]. Agmatine derivatives were also observed in barley, including complex hordatine A, B and C structures that possess antifungal activities [58]. Agmatine conjugates with hydroxycinnamic acids were also reported in wheat [73] and species representing other plant families like African shrub *Maerua edulis* in which *cis-trans* conformers of *p*-coumaroylagmatine were distinct [74].

#### 2.3.4. Flavonoid Glycosides

The highest number of compounds identified in this study belonged to flavonoids. MS^n^ fragmentation schemes of the same or similar molecules were previously described in Poaceae plants including barley and wheat [58,62,75]. Therefore, structural similarities and conservation of structures can be deduced for those closely related species. Our analysis resulted in identification of isomeric structures of glycosylated flavonols (quercetin, isorhamnetin), flavones (apigenin, luteolin and chrysoeriol) and proantocyanidins. The aglycone type of molecular mass 270 and 286 Da were further confirmed as apigenin and luteolin, respectively, referring to pseudo-MS3 spectrum described previously [76]. The characteristic product ions at *m*/*z* 199, 175, 151 and 133 determined the aglycone as luteolin whereas the characteristic product ions at *m*/*z* 153, 145, 121 and 119 determined the aglycone as apigenin. In roots, derivatives of *O*-methylated flavonoids like isorhamnetin, tricin and chrysoeriol were mainly identified. Glycosides of all these aglycones were present in Poaceae species in diverse structural configuration [58,62,75]. We mainly found hexose(s), deoxyhexose(s) and pentose(s) sugar substituents of analyzed flavonoid aglycones. However, a precise sugar structure could not be determined by MS analysis. The presence of glucose and galactose could be suggested in Brachypodium flavonoid glycoconjugates as both hexoses were found in flavonoids of closely related species barley and wheat [77,78]. Among other sugar only rhamnose as deoxyhexose and arabinose as pentose were described in flavonoids of Poaceae family.

Among the different aglycone-sugar conjugates, we identified *O*-glycosides and *C*-glycosides as well as *O-,C*-glycosides. In addition, LC-MS^n^ analysis of flavonoid diglycosides enabled distinguishing interglycosidic bonds in glycosyl(1→6)glycosides and glycosyl(1→2)glycosides [58,62]. Unfortunately, the exact positions of aglycone substitutions in flavonoid glycosides were difficult to establish without NMR analysis. However, on the basis of similarities in fragmentation scheme with mass spectra of flavone derivatives reported in species closely related to Brachypodium, including barley and wheat, 4-OH- and 7-OH- groups could be suggested as positions of glycosylation of flavonoid derivatives from Brachypodium [58,62,75].

##### *O*-Glycosides

Deprotonated [M−H]^−^ ions of compounds **41**, **52** and **78** underwent sequential losses of two 162 Da fragments, first in the MS2 and then MS3 spectra. The HR LC-MS analysis confirmed detachment of two C_6_H_10_O_5_ fragments, which indicated two dehydrated hexoses moieties [hexose-H_2_O]^−^. Neutral losses of dehydrated glycosidic residues are characteristic for *O*-hexosides of flavonoids [58,62]. During MS^n^ analysis of compounds **41, 52** and **78** detachments of two hexosyl units was observed indicating two separate glycosylation sites at flavonoid aglycone (Table 3). The main product ion at *m*/*z* = 287.05247 Da obtained from [M+H]^+^ ions of **41** and **52** as well as *m*/*z* = 301.07037 Da from [M+H]^+^ of **78** corresponded to luteolin and chrysoeriol, respectively. Therefore, these compounds were assigned as luteolin di-*O*-hexoside (**41** and **52**) and chrysoeriol di-*O*-hexoside (**78**).

Unlike compounds **41, 52** and **78**, metabolites **89** and **91** have the main [M−H-308] fragment typical for *O*-hexosyldeoxyhexoside moiety. The [M+H-162]^+^ and [M+H-146]^+^ ions were detected at a lower abundance, suggesting a deoxyhexosyl(1→6)hexosidic bond in these structures according to [79]. The major protonated product ions at *m*/*z* = 331.08045 and *m*/*z* = 301.07037 Da corresponded to tricin and chrysoeriol aglycones, respectively. A relatively high, intense [M+H-164]^+^ ion in MS2 spectrum of metabolite **90** proved that deoxyhexose is external sugar in this molecule. Furthermore, the detachment of an entire moiety of deoxyhexose was typical for deoxyhexosyl(1→2)glycosyl interglycosidic bond.

##### *O,C*-Glycosides

Compounds **68**, **69** and **76** with [M−H]^−^ at *m*/*z* = 593.15155 and calculated chemical formula C_27_H_30_O_15_ have been identified as isomeric structures of apigenin di-hexoside (Figure 7). Differences in mass spectra of these isomers obtained in negative MS^n^ mode enabled to distinguish 7-*O*, 2″-*O* and 6″-*O*-glycoconjugates according to [58,80]. The major [Agly+42-H]^−^ and [Agly + 72-H]^−^ product ions of compounds **68** and **76** were typical for *C*-glycosides of apigenin (isovitexin) [80]. In addition, the main product [M-162]^−^ ion of **76** indicated that the second hexoside was at the 7-*O*-position in the flavone aglycone, which collectively allowed us to assume that **76** was apigenin 6-*C*,7-*O*-di-hexoside (probably isovitexin 7-*O*-glucoside, common name saponarin) (Figure 7D). The main product ions, [M-90-H]^−^ and [M-102-H]^−^, observed in the mass spectra of compound **68**, were previously reported for 6-*C*-[6″-O-hexoside]-hexoside thus **68** was suggested to have the structure of apigenin 6-*C*-[6″-*O*-hexoside]- hexoside (Figure 7B). The presence of [Agly + (42-18)-H]^−^ and [Agly + (72-18)-H]^−^ suggested a 6-*C*-[2″-*O*-hexoside]- hexoside structure in **69** (Figure 7C).

*O*-glucoside. Postulated structures and proposed simplified fragmentation schemes are shown. The product ions subjected to fragmentation in MS3 or MS4 are indicated with turned squares at the ion apexes.

##### Di-*C*-Glycosides

Compounds **57**, **58**, **72**, **73**, **75** and **79-81** were identified as flavonoid di-*C*-glycosides on the basis of characteristic fragmentation and main product ions [Agly+84-H]^−^ and [Agly+114-H]^−^ [80] (Figure 7A). The identification of compounds **57**, **68**, **69** and **76** highlighted the problem with the automatic annotation of metabolomic signals resulting from the diversity of isomeric or isobaric structures available in mass spectra databases. In the case of these four compounds (*m*/*z* = 593.15155), we found 110 entries (with error ppm = 5) in the Metlin database [81]. For this reason, identification of such isomeric structures could be only done by MS/MS or MS^n^ fragmentation scheme analysis.

Structural similarity in *C*-glycosides of apigenin and luteolin between Brachypodium, barley wheat and maize were significant [58,75,82]. However, *C*-glycosylation could be catalysed by different enzymes in Poaceae plants, as for example, by UDP-glucose-dependent *C*-glucosyltransferase in rice and wheat [83] or by bifunctional *C*-/*O*-glycosyltransferase in maize [82]. In these cereals as well as in Brachypodium, *C*-glycosides of the flavones apigenin and luteolin were dominant metabolites, with glycosylation occurring singly or doubly at the 8-*C* and 6-*C* positions. Structural isomerism related to glycoconjugation hindered proper *C*-glycosides identification in Brachypodium plant by simple annotation to dedicated databases. Furthermore, significant changes in content of saponarin isomers can be observed during plant development as was detected for barley [84]; therefore, detailed flavonoids studies in Brachypodium should be further extended in developmental context.

#### 2.3.5. Acylated Flavonoids

Acylated flavones, in which glycosides are substituted with hydroxycinnamic acids, i.e., *p*-coumaric, caffeic, ferulic and sinapic acids, present in barley and wheat have been studied by NMR and mass spectrometry and were identified as 7-*O*-[6″-acyl]-glucosides and 7-*O*-[6″-acyl]-glucosyl-4’-*O*-glycosides [85,86,87]. Surprisingly, in Brachypodium we identified only one acylated flavonoid (**60**) that was detected in leaves while in Poaceae family different isomeric and isobaric structures are present. According to literature data from other related species the structure was established as isoorientin 2″-*O*-hexoside 7-*O*-[6″-sinapoyl]-hexoside [58]. The absence of acylated flavonoids in our analysis may eventually arise from our plant growth conditions and/or age of plants used in our analysis, therefore presence of acylated flavonoids in Brachypodium cannot be excluded.

#### 2.3.6. Flavan-3-ols

Our LC/MS analysis revealed that spikes of Brachypodium were rich in flavan-3-ol derivatives, mainly from the proanthocyanidin group. Stereoisomers catechin and epicatechin as well as gallocatechin and epigallocatechin showed the same product ions and very similar ratios on the MS/MS or MS^n^ spectra, thus even preliminary identification of these isomers was not possible based on the obtained MS spectra. The position and the stereochemistry of the interflavan linkage could not be elucidated by MS. Procyanidins can be classified into A-type and B-type depending on the stereo configuration and linkage between monomers. B-type procyanidins possess a single C-C interflavan bond while A-type procyanidins have additional ether bond. The (epi)catechin, (epi)gallocatechin and galloyl subunits were identified in Brachypodium as procyanidin A- and B-type on the basis of typical fragment detachments in MS/MS of deprotonated molecules according to [88,89]. Compounds **24** and **25** have the main product ions at *m*/*z* = 305.07 and 289.07 Da indicating (epi)gallocatechin as a core aglycone. The observed main fragments detached from the [M−H]^−^ ion of **24** and **25** at 120.021 and 152.011 Da, corresponding to chemical formulae and masses of hydroxybenzoic and gallic acids, respectively. Therefore, **24** and **25** were identified as (epi)gallocatechin *O*-hydroxybenzoate and *O*-gallate.

High resolution mass spectrometry as well as MS^n^ enabled to trace the way of fragmentation of dimeric and trimeric structures of (epi)catechin and (epi)gallocatechin proanthocyanidin. An example of the fragmentation of compound **15** is the hRetro Diels-Adler reaction (RDA), which resulted in the detachment of fragment 168 Da, typical for (epi)gallocatechin oligomers (Figure 8). The most abundant product ion at *m*/*z* = 407 Da resulted from water elimination from RDA product and [M−H-126]^−^ ions from heterocyclic ring fission (HRF). In the parallel fragmentation, a Quinone-Methide cleavage of interflavan bond occurred and the remaining deprotonated ions corresponded to (epi)catechin monomer. Therefore, **15** was identified as a procyanidin B-type dimer. Compounds **7** and **27** possessed a similar fragmentation scheme as compound **15**; however, their m/z ratios indicated A-type interflavan bonds in those proanthocyanidis. Trimeric proanthocyanidins (**6**, **7** and **20**) were also observed. The typical RDA and QM fragments corresponded to (epi)catechin and (epi)gallocatechin subunits in those structures.

Proanthocyanidins are well described in several cereal and grass species, including sorghum and barley, due to their nutritional and technological importance [90]. For instance, barley grains have been shown to be especially rich in proanthocyanidins, which contribute to haze formation in barley beer [91]. However, despite well-characterized Brachypodium genes responsible for catechin and epicatechin biosynthesis [92] the number of studies on the accumulation of proanthocyanidins in Brachypodium plants is very limited. Only (-)-epicatechin was detected by widely targeted metabolome analysis in seeds and leaves of Bd21 and Bd3-1 [12]. Overall, our study revealed differential accumulation of proanthocyanidins among Brachypodium organs. We observed high accumulation levels of these compounds in seeds and intermediate accumulation level in leaves while only compounds **6** ((epi)gallocatechin trimer) and **24** ((epi)gallocatechin *O*-hydroxybenzoate) were detected in roots. Di- to penta-mers and galloylated proanthocyanidins were found in barley. Epiafzelechin derivatives, which are abundant in buckwheat (*Fagopyrum esculentum)* [93], were not detected during our analysis.

## 3. Material and Methods

### 3.1. Plant Material

Seeds of Brachypodium lines Bd21 and Bd3-1 were obtained from Robert Hasterok (University of Silesia, Katowice, Poland). Plants were grown in soil in a controlled growth chamber at 23 or 20 °C (day or night) under short day conditions (8 h light and 16 h darkness) for 4 weeks at 50–60% relative humidity and irradiance of 100 µmol m^−2^ s^−1^. Seeds were sown in 9 × 9 cm square pots filled with soil mixed with peat. In the third week of cultivation a supplemental liquid fertilizer (N total 12% *w*/*w*, P_2_O_5_ 4% *w*/*w*, K_2_O 6% *w*/*w*, B 0.01% *w*/*w*, Cu 0.0007% *w*/*w*, Fe 0.015% *w*/*w*, Mn 0.012% *w*/*w*, Mo 0.001% *w*/*w*, Zn 0.005% *w*/*w*) was applied in a concentration of 1 mL/1 L of water. Subsequently, light conditions were changed to a long day (16 h light and 8 h darkness) and plants were grown until the developmental stage of spikes, i.e, watery ripe: first grains have reached half their final size (71–73 in BBCH scale according to [94]). Samples of the spikes, leaves and roots were collected, weighted and placed in 2ml tubes containing zirconia beads, then immediately frozen in liquid nitrogen and stored at −80 °C until further processing. An extraction buffer containing 0.5 mM lidocaine and 0.5 mM camphorsulfonic acid in DMSO was added (2.5 µL /1 mg of fresh plant weight) to each tube. Samples were homogenized with a Precellys Evolution (Bertin, France) tissue grinder and centrifuged in 4 °C at 15,000 g. Supernatants were collected and directly subjected to LC-MS analysis.

### 3.2. Chemicals

The lidocaine and camphorsulfonic acid were from Merck SA (Darmstadt, Germany and the DMSO from Bioshop (Burlington, ON, Canada). Acetonitrile for LC-MS analyses was from VWR Chemicals (Radnor, PA, USA) and the formic acid was from Merck SA (Darmstadt, Germany). Standards of compounds (caffeic acid, isoorientin, apigenin 6-*C*-glucoside-8-*C*-arabinoside, luteolin-3,7-di-*O*-glucoside, tricin 7-*O*-glucoside and tricin glucosylrhamnoside) were purchased from Extrasynthese (Genay, France). Isoorientin 2″-*O*-glucoside, isovitexin 7-*O*-glucoside, isoscoparin 2″-*O*-glucoside and apigenin 6-*C*-arabinoside-8-*C*-glucoside were purified from plant material and their structures were confirmed with NMR analysis as described previously [58].

### 3.3. Metabolite Profiling

Analysis and identification of metabolites was performed using two complementary LC-MS systems using the previously published approach [58]. First of them (low resolution HPLC-DAD-MS^n^) consisted of 1100 HPLC system with a photodiode-array detector (Agilent, Santa Clara, CA, USA) equipped with an XBridge Shield C18 column (150 × 2.1 mm, 3.5 μm particle size; (Waters, Milford, CT, USA) coupled to an Esquire 3000 ion trap mass spectrometer (Bruker Daltonics, Billerica, MA, USA). Chromatographic separations were conducted with injection volume 10 µL using water with 0.1% formic acid (solvent A) and acetonitrile (solvent B) at the flow rate 0.2 mL/min and the following gradient: 0–6 min from 8% to 10% B, 6–40 min to 20% B, 40–46 min to 98% B maintained for 5 min. The MS^n^ spectra were separately recorded in the negative and positive ion modes. The second system (high resolution UPLC-MS/MS) consisted of UPLC equipped with a photodiode-array detector (Acquity System; Waters) hyphenated to a high-resolution Q-Exactive hybrid MS/MS quadrupole Orbitrap mass spectrometer (Thermo Fisher Scientific, Waltham, MA, USA). Chromatographic separation was performed on an Acquity UPLC HSS T3 C18 chromatographic column (2.1 × 50 mm, 1.8 μm particle size; Waters) at 22 °C using water containing 0.1% formic acid (solvent A) and acetonitrile (solvent B). The injection volume was 5 μL, gradient elution started at 100% of A and linearly changed to 20% of B over 2 min, then to 30% of B over 8 min and to 95% of B over 1 min maintained for 2 min. UV absorbance was recorded in the 230–450 nm wavelength range with a resolution of 2 nm. Q-Exactive MS operated in Xcalibur version 3.0.63 with the following settings: heated electrospray ionization ion source voltage −3 kV or 3 kV; sheath gas flow 30 L/min; auxiliary gas flow 13 L/min; ion source capillary temperature 250 °C; auxiliary gas heater temperature 380 °C. MS/MS mode (data-dependent acquisition) was recorded in negative and positive ionization, at resolution 70,000 and AGC (ion population) target 3e6, scan range 80 to 1000 *m/z.*

### 3.4. Metabolite Identification

Individual compounds were tentatively identified on the basis of low resolution LC-MS^n^ mass spectra if corresponding fragmentation spectra and *m*/*z* signals were confirmed in high resolution UPLC-MS/MS. Particular structures were suggested via comparison of the exact molecular masses with ∆ less than 5 ppm, mass spectra and retention times to those of standard compounds, spectra in available databases (PubChem, ChEBI, Metlin, Reaxys, DynLib and KNApSAck) [48,66,81,95,96,97] and literature data. Confirmation of isomeric aglycone type was based on available standard compounds and methods described previously [76]. Pseudo-MS3 spectra of flavonoids *O*-glycosides with in-source CID 80 eV enabled to confirm fragmentation typical for luteolin based on the presence of product ions at *m*/*z* 117 and 135 for and excluded presence of other isomers e.g., kaempferol.

### 3.5. Bioinformatic Processing

High-resolution raw UPLC-MS/MS data were separately processed by MZmine 2.53 [98] for negative and positive ionization. In first step, lists of masses were generated in each scan of the raw data files (Appendix A). Chromatograms for each exact mass detected over the scans were built by a Chromatogram Builder algorithm. These chromatograms were deconvoluted using an ADAP Wavelets algorithm and subsequently subjected to isotope elimination, adduct and complex searching, followed by retention times normalization among peak lists. Such transformed peaks were aligned across all samples by a Join aligner module. The resulting peak table was completed by supplemental peak detection with a peak finder algorithm prior to missing value imputation (gap-filling). The obtained result table was subjected to further statistical analysis and visualizations.

### 3.6. Statistical Analysis

Statistical analysis was performed with a Genstat 21 (VSN International, Hempstead United Kingdom). Observations below the detection limit were substituted with half of the minimum non-zero observation for each metabolite and then observations were transformed by log_2_(x). Two-way analysis of variance (ANOVA) was performed with the experiments as a block (random effects) and organ line as two fixed factors. Analysis was performed together with positive and negative ionization. Significant changes in the accumulation of metabolites was indicated by the effect on an organ, line or by the interaction an organ x line with *p*-value < 0.05. To select metabolites with significant differences between lines in each organ, the definition of DAM was introduced. Only signals with a significant interaction of line and organ (LxO) or with significant effects on a line (L) were classified as DAMs. Additionally, for each signal, we calculated fold change (FC) in each organ as a ratio of signal intensities in the Bd3-1 and Bd21 lines (Bd3-1/Bd21) to restrict analysed DAMs to those with FC > 2 (|log_2_(Bd3-1/Bd21)| > 1). Visualizations including PCA 3D plot (generated with data after log_2_ transformation), heatmaps and Venn diagrams were created in R (R Foundation for Statistical Computing).

### 3.7. Pathway Enrichment Analyses

Pathway enrichment analyses were conducted with all *m*/*z* signals from the combined positive and negative ionization modes and only with *m*/*z* signals representing DAMs. Data was imported to a functional analysis module in MetaboAnalyst 5.0 [40]. This enabled direct *m*/*z* value annotation to metabolic data base for *O. sativa* subsp. *Japanese* from the Kyoto Encyclopedia of Genes and Genomes (KEGG) [35,37]. Signals with significant annotation were further subjected to pathway-level enrichment on the basis of mummichog algorithms in a pathway analysis module of MetaboAnalyst 5.0. This module filtered metabolites over-represented on the pathway level in addition to pathway topology analysis. Enrichment was ranked on the basis of mummichog algorithms followed by Benjamini-Hochberg false discovery rate (FDR) correction. Pathway topology was scored on the basis of relative-betweenness centrality, to estimate the relative importance of individual nodes to the overall pathway network. The node impact values were normalized by the sum of the importance of the pathway to estimate maximum impact of each pathway as 1. Significantly enriched metabolic pathways upon differentiating factors were selected if FDR < 0.03 and pathway impact > 0.3 were consistent across multiple comparisons.

## Figures and Tables

**Figure 1 molecules-27-05956-f001:**
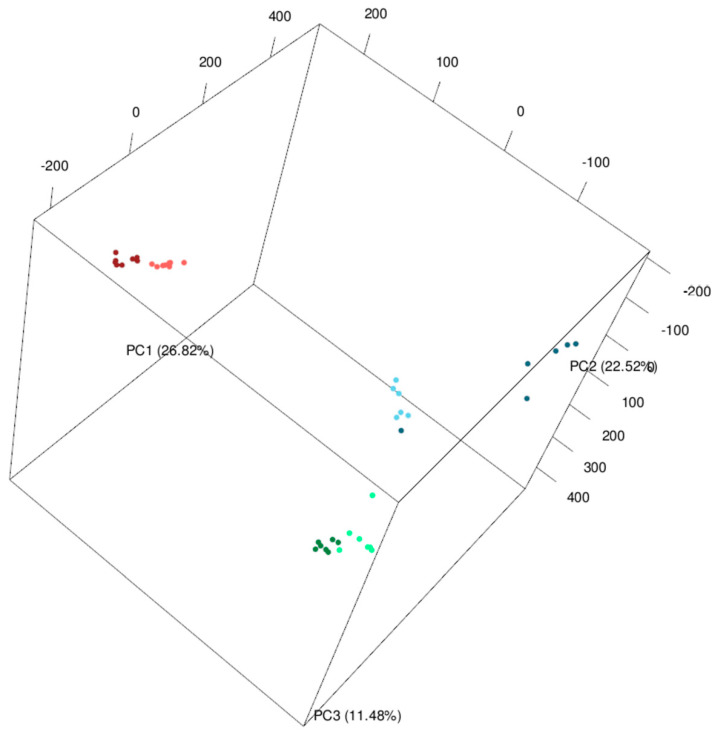
Three-dimensional principal component analysis (PCA) plot of global metabolite profiles in leaves (green), spikes (blue) and roots (red) of Brachypodium Bd21 (light colours) and Bd3-1 (dark colours) lines.

**Figure 2 molecules-27-05956-f002:**
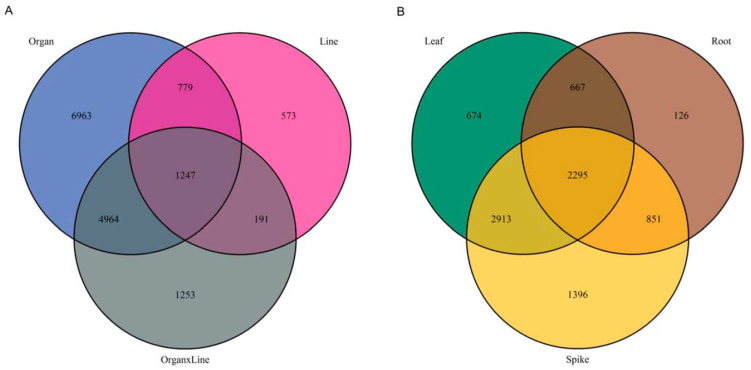
Venn diagrams indicating (**A**) number of shared and unique signals (*p*-value < 0.01) in Brachypodium with significant effect of organ (O), line (L), or interaction organ × line (O × L); (**B**) number of shared and organ-specific differentially accumulating metabolites (DAMs) defined as signals meeting the conditions: *p*-value < 0.05 for factor L or O × L; |log_2_ (fold change)| > 1.5, where fold change was Bd3-1/Bd21 signal intensities.

**Figure 4 molecules-27-05956-f004:**
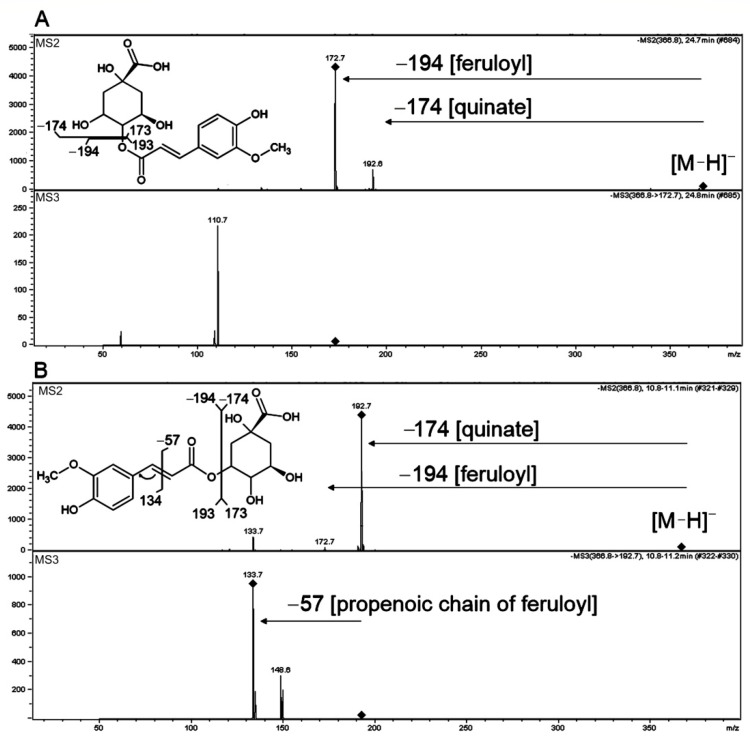
Low resolution MS2 and MS3 fragmentation spectra obtained in negative ionization of isomeric hydroxycinnamoyl-quinic acid (HQA) conjugates with hydroxycinnamic acids. (**A**) compound **43**, 4-feruloylquinic acid, (**B**) compound **26**, 5-feruloylqiunic acid. Postulated structures and proposed simplified fragmentation schemes are shown. The product ions subjected to fragmentation in MS3 or MS4 are indicated with turned squares at the ion apexes.

**Figure 5 molecules-27-05956-f005:**
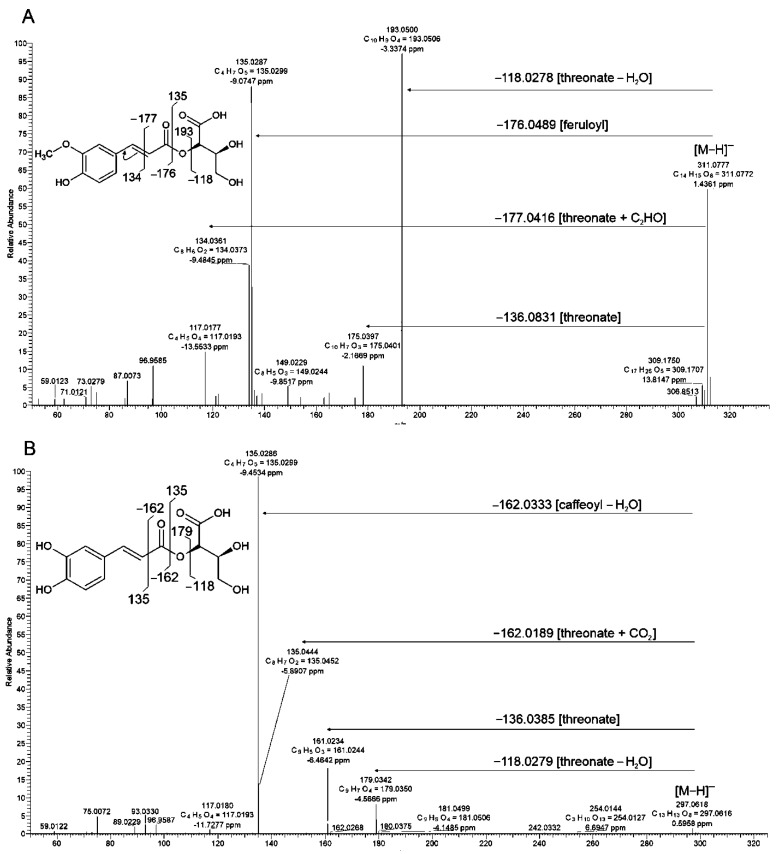
High resolution MS/MS spectra with simplified fragmentation schemes obtained in negative ionization. (**A**) compound **42**, feruloylthreonic acid, and (**B**) compound **18**, caffeoylthreonic acid. Postulated structures are shown as 2-*O*-threonate, however, precise esterification position on threonic acid cannot be solved by MS fragmentation analysis.

**Figure 6 molecules-27-05956-f006:**
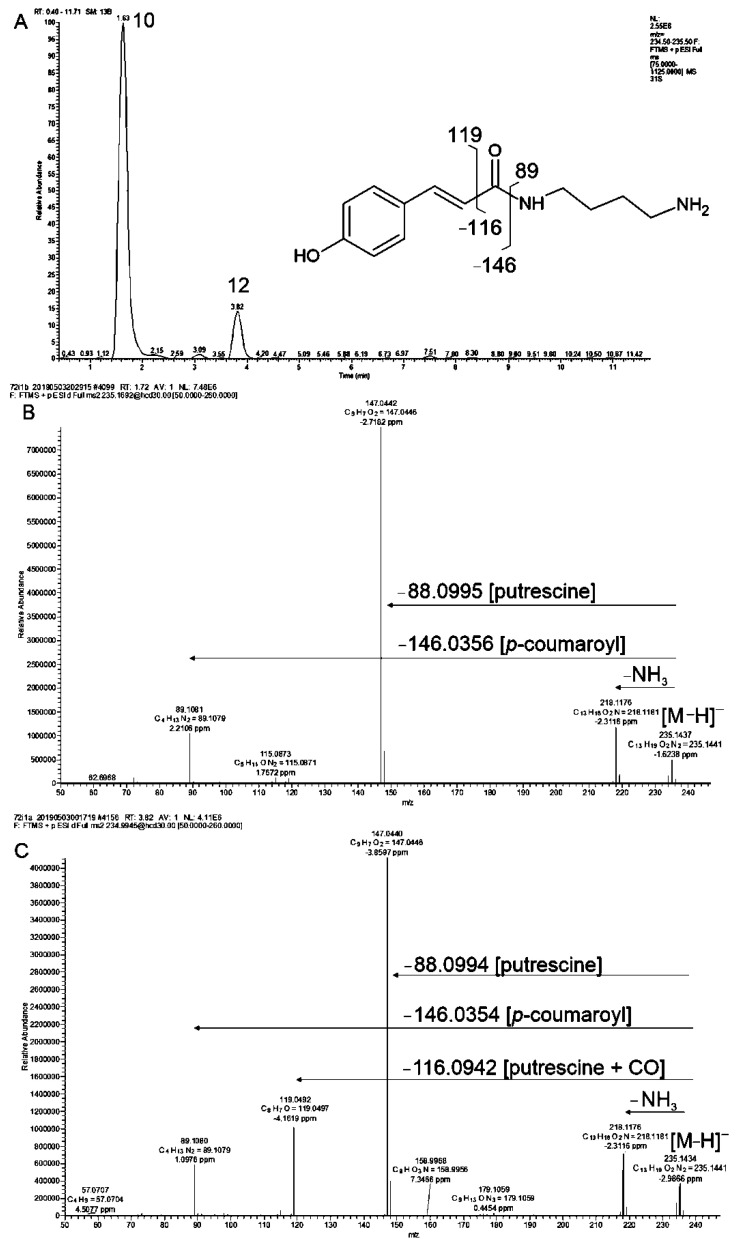
Representative chromatogram (**A**) and high-resolution MS/MS spectra obtained in positive ionization of two isomeric structures of *p*-coumaroyl-*N*-putrescine (**B**) compound **10** and (**C**) compound **12**. Postulated structure and proposed simplified fragmentation schemes are shown.

**Figure 7 molecules-27-05956-f007:**
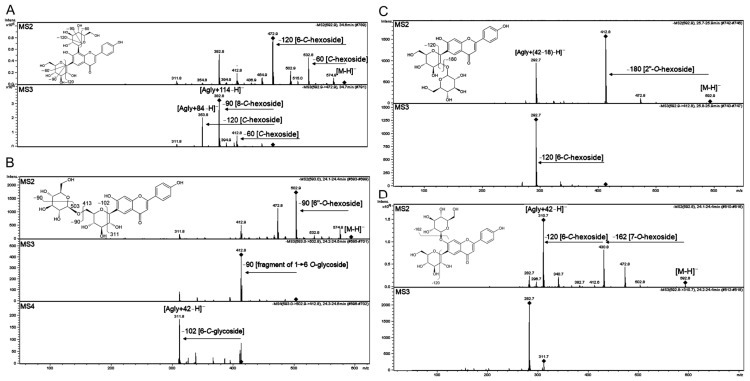
Identification of isomeric structures of apigenin di-glucosides in Brachypodium based on low-resolution MS2, MS3 and MS4 fragmentation spectra obtained in negative ionization. (**A**) Compound **57**, apigenin 6,8-di-*C*-glucoside, (**B**) compound **68**, apigenin 6-*C*-[6″-*O*-glucoside]-glucoside, (**C**) compound **69**, apigenin 6-*C*-[2″-*O*-glucoside]-glucoside, (**D**) compound **79** isovitexin 7.

**Figure 8 molecules-27-05956-f008:**
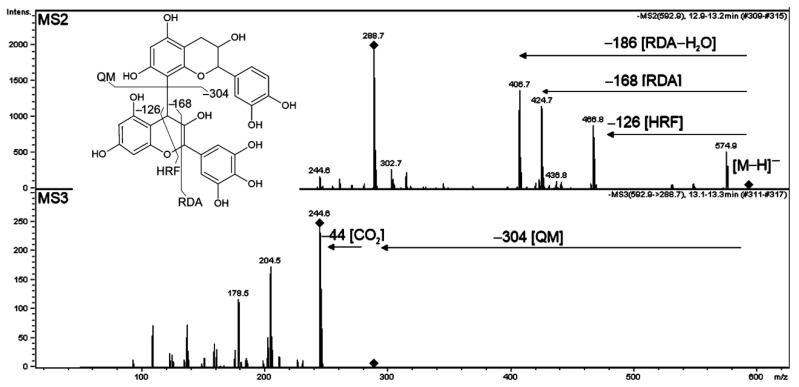
Low-resolution MS2 and MS3 fragmentation spectra obtained in negative ionization supporting identification of compound **15**, prodelphinidin B-type structure. Postulated structure and proposed simplified fragmentation scheme are shown. The product ions subjected to fragmentation in MS3 or MS4 are indicated with turned squares at the ion apexes.

**Table 1 molecules-27-05956-t001:** The most represented metabolic pathways based on pathway enrichment analysis performed with all MS signals detected in Brachypodium roots, leaves and spikes. KEGG—Kyoto Encyclopedia of Genes and Genomes [37]; total—number of compounds included in biological pathway in the database; hits—number of compounds matched in our analysis; FDR—false discovery rate; impact—pathway impact value related to the number of links occurred upon a node in pathway topology graph. Full set of data including other pathways as well as lists of annotated metabolites is available as Appendix A.

Biological Pathway (KEGG)	Total	Hits	FDR	Impact
Flavonoid biosynthesis	47	40	4.31 × 10^−6^	0.75288
Galactose metabolism	27	25	2.72 × 10^−5^	1
Amino sugar and nucleotide sugar metabolism	50	39	0.000177	0.92767
Valine, leucine and isoleucine biosynthesis	22	20	0.000426	0.99998
Pentose and glucuronate interconversions	17	16	0.000971	1
Flavone and flavonol biosynthesis	12	12	0.001539	1
2-Oxocarboxylic acid metabolism	12	12	0.001539	1
Pentose phosphate pathway	19	16	0.010995	0.9532
Purine metabolism	63	42	0.011504	0.67941
Tyrosine metabolism	18	15	0.014584	0.79191
Ascorbate and aldarate metabolism	18	15	0.014584	0.8806
Vitamin B6 metabolism	11	10	0.026026	0.96153

**Table 2 molecules-27-05956-t002:** Metabolic pathways discriminating Bd21 and Bd3-1 lines in particular organs selected based on pathway enrichment analysis performed with MS signals representing differentially accumulating metabolites (DAMs). KEGG—Kyoto Encyclopedia of Genes and Genomes [37]; total—number of compounds included in biological pathway in the database; hits—number of compounds matched in our analysis; FDR—false discovery rate; impact—pathway impact value related to the number of links occurred upon a node in pathway topology graph. Full sets of data including other pathways as well as lists of annotated metabolites are available as Appendix A.

	Biological Pathway Enrichment (KEGG)	Total	Hits	FDR	Impact
**Roots**	Galactose metabolism	27	25	7.41 × 10^−7^	1
	Diterpenoid biosynthesis	47	32	0.001253	0.61959
	Flavonoid biosynthesis	47	32	0.001253	0.64212
	Valine, leucine and isoleucine biosynthesis	22	18	0.001253	0.68294
	Caffeine metabolism	10	10	0.001636	0
	Pentose phosphate pathway	19	15	0.007666	0.85473
	One carbon pool by folate	8	8	0.007666	1
	Flavone and flavonol biosynthesis	12	10	0.025063	0
	2-Oxocarboxylic acid metabolism	12	10	0.025063	0
**Leaves**	Flavonoid biosynthesis	47	42	2.39 × 10^−6^	0.7644
	Galactose metabolism	27	24	0.001979	1
	Valine, leucine and isoleucine biosynthesis	22	20	0.003065	0.8355
	Histidine metabolism	17	16	0.004891	1
	Flavone and flavonol biosynthesis	12	12	0.0054	0
	2-Oxocarboxylic acid metabolism	12	12	0.0054	0
	Pentose phosphate pathway	19	17	0.007434	0.99999
	Vitamin B6 metabolism	11	11	0.007908	0.99999
	Diterpenoid biosynthesis	47	34	0.028047	0.69919
**Spikes**	Flavonoid biosynthesis	47	41	3.39 × 10^−5^	0.75288
	Galactose metabolism	27	25	0.000367	1
	Pentose phosphate pathway	19	18	0.002628	0.99999
	Valine, leucine and isoleucine biosynthesis	22	20	0.003264	0.8355
	Flavone and flavonol biosynthesis	12	12	0.006808	0
	2-Oxocarboxylic acid metabolism	12	12	0.006808	0
	Vitamin B6 metabolism	11	11	0.011173	0.99999
	Pentose and glucuronate interconversions	17	15	0.021023	0.85716
	Histidine metabolism	17	15	0.021023	1
	Cysteine and methionine metabolism	46	34	0.021023	0.75798

**Table 3 molecules-27-05956-t003:** Specialized metabolites identified in leaves, roots and spikes of Brachypodium, using two complementary MS systems: HPLC-ESI-MS^n^ and UPLC-HR-MS/MS. Chemical formulas were calculated on the basis of accurate masses measured in HR-MS/MS, and fragmentation pathways are given on the basis of ESI-MS^n^. The main peaks in MS2 or MS3 taken for further fragmentation are highlighted in bold. Identification levels are given according to the Metabolomics Standards Initiative recommendation [47]. ChEBI—respective identifiers of chemical structure in the Chemical Entities of Biological Interest database [48]. *—indicates detection of metabolites in particular organs. #—ChEBI identifiers for other optic isomers of the compound: 75667, 75666, 75668, 75672, 75670, 75669. Std.—identification supported with analysis of available standard compounds; sh—spectrum shoulder.

#	Fragmentation Pathway in MS^n^ [*m*/*z*]		Identification	Exact mass of [M+H]^+^ or [M−H]; [Da]			∆ ppm	Chemical Formula	λ_max_ [nm]	Leaves	Roots	Spikes	ChEBI	Identification Level	References
	Negative Ionization	Positive Ionization		Ion Type	Measured	Calculated									
1		**MS2**: 137, 90, 64	Dopamine	[M+H]^+^	154.08638	154.0864	0.8102	C_8_H_11_NO_2_		*	*	*	18243	2	[49]
2		**MS2**: **165**, 147, 136 **MS3**: 147, 123	Tyrosine	[M+H]^+^	182.081	182.0812	−1.1348	C_9_H_11_NO_3_		*	*		18186	2	[50]
3		**MS2**: 116, 86	Leucine (Isoleucine)	[M+H]^+^	132.1018	132.1019	−0.5152	C_6_H_13_NO_2_		*	*	*	25017	2	[50]
4		**MS2**: 163, 89 **MS3**: 131	*N*-Caffeoyl-putrescine	[M+H]^+^	251.13862	251.139	−1.589	C_13_H_18_N_2_O_3_		*	*	*	17417	3	[51]
5		**MS2**: **145**, 120 MS3: 79	Phenylalanine	[M+H]^+^	166.086	166.0863	−1.3614	C_9_H_11_NO_2_	260	*	*	*	28044	2	[50]
6	**MS2**: **727**, 609, 559, 541, 483, 423, 303 **MS3**: 559, 423, 303		(epi)Gallocatechin trimer	[M−H]^−^	913.18583	913.1833	2.791	C_45_H_38_O_21_		*	*	*		3	[52]
7	**MS2**: 771, 711, 593, 543, 467, 303, 289 **MS3**: 697, 543, 289		Proanthocyanidins trimer A-type	[M−H]^−^	897.19086	897.1884	2.779	C_45_H_38_O_20_				*		3	[52]
8	**MS2**: 305, 265, 223, 205, 161, 143, 125 **MS3**: 223, 205		Caffeic acid derivative	[M−H]^−^	367.12504	367.1246	1.24	C_17_H_20_O_9_		*			149782	2	[53]
9	**MS2**: 269, 209, 167		Vanilic acid-hexoside	[M−H]^−^	329.0883	329.0878	1.5932	C_14_H_18_O_9_		*		*		2	[54]
10		**MS2**: **218**, 89	*p*-Coumaroyl-*N*-putrescine	[M+H]^+^	235.1441	235.1441	0.0370	C_13_H_18_*N*_2_O	290sh	*	*		70431	2	[51]
11		**MS2**: 248, **177**, 144, 114, 98 **MS3**: 145	Feruloyl-*N*-putrescine	[M+H]^+^	265.15424	265.1547	−1.16337	C_14_H_20_N_2_O_3_		*	*		9299	3	[51]
12	**MS2**: 233, 119 **MS3**: 117, 93	**MS2**: **218**, 176, 147, 114, 89, 73 **MS3**: 147	*p*-Coumaroyl-*N*-putrescine	[M+H]^+^	235.14375	235.1441	−1.507	C_13_H_18_*N*_2_O		*	*			3	[51]
13	**MS2**: 203, 159, 142, 116	**MS2**: **188**, 146 **MS3**: 146, 118	Tryptophan	[M+H]^+^	205.097	205.0972	−0.9744	C_11_H_12_N_2_O_2_	285	*	*	*	27897	1	Std
14	**MS2**: 299, 239, **209**, 179, 137 MS3:		Hydroxybenzoic acid hexoside	[M−H]^−^	299.0765	299.07728	−2.349	C_13_H_16_O_8_	282			*	16741	2	[54]
15	**MS2**: 574, 467, **425**, 407, 289 **MS3**: 245, 205, 177	**MS2**: 595, 443, **427**, 317, 307, 289 **MS3**: 289, 247	Prodelphinidin B-type	[M−H]^−^	593.13153	593.1301	2.472	C_30_H_26_O_13_				*	75664#	2	[52,55]
16	**MS2**: 315, **153** **MS3**: 108		Dihydroxybenzoic acid hexoside	[M−H]^−^	315.0718	315.0722	−1.1026	C_13_H_16_O_9_	286			*		2	[54]
17		**MS2**: **160** **MS3**: 134, 132, 115	Serotonin	[M+H]^+^	177.1019	177.1022	−2.0138	C₁₀H₁₂N₂O	275, 298sh	*		*	28790	1	Std
18	**MS2**: 179, **135** **MS3**:117, 89, 75	**MS2**: 163, **136**,137 **MS3**: 136, 118	Caffeoylthreonic acid	[M−H]^−^	297.0611	297.0616	−1.664	C_13_H_14_O_8_		*	*	*		2	[54]
19	**MS2**: 461, 225, **153** MS3: 108, 90		Dihydroxybenzoic acid hexosyldeoxyhexoside	[M−H]^−^	461.1299	461.1301	−0.459	C_19_H_26_O_13_	281			*		3	[54]
20	**MS2**: 863, 755, **695**, 591, 407, 289, 243 MS3: 524, 283		Catechin-gallocatechin-catechin	[M−H]^−^	881.19622	881.1935	3.141	C_45_H_38_O_19_				*		3	[52,55]
21	**MS2**: 179, 134, 119		Caffeic acid	[M−H]^−^	179.03439	179.035	−3.307	C_9_H_8_O_4_	304sh	*	*		36281	1	Std
22	**MS2**: 299, 239, **197**, 153, 138 **MS3**: 182, 153,138, 121		Syringic acid-hexoside	[M−H]^−^	359.09827	359.09782	1.2393	C_15_H_20_O_10_		*		*		2	[54]
23		**MS2**: **248**, 177, 145 MS3: 177	feruloyl-*N*-putrescine	[M+H]^+^	265.1548	265.1547	0.3229	C_14_H_20_N_2_O_3_		*	*			3	[51]
24	**MS2**: 305, 289, **241**, 225, 139 **MS3**: 223, 184, 139, 97	**MS2**: 337, 305, **185**, 153 **MS3**: 153, 125	(epi)Gallocatechin *O*-hydroxybenzoate	[M−H]^−^	425.08831	425.0878	1.187	C_22_H_18_O_9_		*	*	*		3	[52,55]
25	**MS2**: 439, 325, 305, 289, 191, **163**, 131 **MS3**: 115		(epi)Gallocatechin 3-*O*-gallate	[M−H]^−^	457.07885	457.0776	3.859	C_22_H_18_O_11_		*				`3	[52,55]
26	**MS2**: 323, **193**, 173, 135 **MS3**: 149, 135	**MS2**: 353, 309, 274, 238, **177**, 145 **MS3**: 145	5-Feruloylquinic acid	[M−H]^−^	367.1028	367.1035	−1.831	C_17_H_20_O_9_	280sh, 320	*		*	86388#	2	[56]
27	**MS2**: 607, **589**, 333, 203 **MS3**: 333, 203		Prodelphinidin A-type dimer (Prodelphinidin A1)	[M−H]^−^	607.11102	607.1093	2.786	C_30_H_24_O_14_				*			[52,55]
28	**MS2**: 463, 301	**MS2**: 465, 303, 229, 201 **MS3**: 303	Quercetin di-*O*-hexoside	[M−H]^−^	625.14292	625.141	3.035	C_27_H_30_O_17_	253, 353	*				3	[53,57]
29	**MS2**: 353, 179, **173** **MS3**: 109, 93	**MS2**: 192, 165, 146	4-Caffeoylquinic acid	[M−H]^−^	353.06743	353.0667	2.136	C_16_H_18_O_9_	340sh, 305	*		*	75491	2	[56]
30	**MS2**: 665, **635**, 563, 503, 473, 443, 383, 353 **MS3**: 353, 297		Apigenin 6-*C*-hexoside-8-*C*-pentoside 7-*O*-hexoside	[M−H]^−^	725.1924	725.1898	−3.4967	C_32_H_38_O_19_	266, 335					3	[58]
31	**MS2**: **489**, 399, 369 **MS3**: 369		Luteolin 6,8-di-*C*-hexoside	[M−H]^−^	609.1435	609.1450	−2.5141	C_27_H_30_O_16_	262, 345	*		*	6553	2	[58]
32	**MS2**: 577, **407**, 289 **MS3**: 289, 143	**MS2**: 579, 427, 291	Procyanidin B-type dimer	[M−H]^−^	577.13679	577.1351	2.843	C_30_H_26_O_12_				*	75630	2	[52,55]
33	**MS3**: **245**, 205, 137, 125 MS3: 203	**MS2**: 157, 139, 123	(epi)Catechin	[M−H]^−^	289.07255	289.0718	2.728	C_15_H_14_O_6_				*	23053	2	[55]
34	**MS2**:595, **483**, 423, 305, 283 **MS3** (609-483): 303, 179 **MS3** (609-305): 289, 143	**MS2**: 611, 443, **317**	Prodelphinidin B-type	[M−H]^−^	609.12659	609.125	2.646	C_30_H_26_O_14_				*		2	[52]
35		**MS2**: **275**, 235, 218, 147, 118 **MS3**: 218, 147, 112	*N-p*-Coumaroyl spermidine	[M+H]^+^	292.07275	292.0717	3.706	C_34_H_37_N_3_O_6_	285sh			*		2	[59]
36	**MS2**: **609**, 301 **MS3**: 301, 272	**MS2**: 627, 611, **465**, 303 **MS3**: 369, 303	Quercetin *O*-deoxyhexosylhexoside-*O*-hexoside	[M−H]^−^	771.1991	771.189	0.1549	C_33_H_40_O_21_	255, 353	*				3	[58]
37	**MS2**: 463, 301	**MS2**: 610, 551, 465, 303 **MS3**: 303	Quercetin di-*O*-hexoside II	[M−H]^−^	625.13971	625.141	−2.099	C_27_H_30_O_17_	253, 353	*	*			3	[53,57]
38	**MS2**: 193, 134		Ferulic acid	[M−H]^−^	193.05024	193.0506	−2.031	C_10_H_10_O_4_	300sh, 326	*	*		17620	3	[53,57]
39	**MS2**: 179, **135**, 117 **MS3**:117, 89	**MS2**: 299, 136	Apigenin 7,4’-dimethyl ether	[M−H]^−^	297.07752	297.0768	2.266	C_17_H_14_O_5_		*			17620	2	[60]
40	**MS2**: 233, 119	**MS2**: **260**, 217, 147, 114 **MS3**: 217, 98	*p*-Coumaroylagmatine	[M+H]^+^	277.16547	277.1659	−1.560	C_14_H_20_N_4_O_2_	295sh	*	*	*	32818	2	[54]
41	**MS2**: 489, **447**, 285, 254 **MS3**: 285, 254	**MS2**: 567, **449**, 287	Luteolin di-*O*-hexoside	[M−H]^−^	609.1477	609.1461	2.614	C_27_H_30_O_16_	267, 348		*			2	[61]
42	**MS2**: 311, **193**, 149, 135 **MS3**: 135, 119		Feruloylthreonic acid	[M−H]^−^	311.07697	311.0772	−0.87	C_14_H_16_O_8_		*				3	[57]
43	**MS2**: 193, **173**, **MS3**: 109, 93	**MS2**: 404, 369, 277, 193	4-Feruloylquinic acid	[M−H]^−^	367.1037	367.1035	0.7349	C_17_H_20_O_9_		*				3	[54]
44	**MS2**: 562, 519, **477**, 315 **MS3**: 357, 315, 285, 243, 199	**MS2**: 641, 479, 317, 286	Isorhamnetin di-*O*-hexoside	[M−H]^−^	639.15509	639.1567	−2.476	C_28_H_32_O_17_	259, 369	*		*	60078	2	[56]
45	**MS2**: 695, 635, **593**, 454, 473, 413, 311, 249 **MS3**: 473, 413		Chrysoeriol 6-*C*-hexoside-8-*C*-pentoside 7-*O*-hexoside	[M−H]^−^	755.20585	755.204	2.428	C_33_H_40_O_20_	250, 348	*	*			3	[57]
46	**MS2**: 519, 447, **357**, 327	**MS2:** 532, 464, **449**, 431, 383, 353, 329, 299 **MS3**: 432, 413, 383, 353, 329, 320, 299	Orientin 7-*O*-hexoside	[M−H]^−^	609.1439	609.1450	−1.8549	C_27_H_30_O_16_	266, 349	*				3	[58]
47	**MS2**: 369, 325, **163**, 145, 119 **MS3**: 117, 95		Sinapoyl-homovanillic acid	[M−H]^−^	387.10814	387.1085	−1.035	C_20_H_20_O_8_				*		3	[58]
48	**MS2**: 191, 179, **173**, **MS3**: 155, 111, 93, 71	**MS2**: 146, 119, 79	4-*p*-Coumaroylquinic acid	[M−H]^−^	337.0939	337.0929	2.6396	C_16_H_18_O_8_	290sh	*	*		1945	3	[54]
49		**MS2**: **290**, 247, 232, 177, 152, 145, 114 **MS3**: 273, **247**, 230, 177, 115 **MS4**: 145, 113	Feruloylagmatine	[M+H]^+^	307.17636	307.1765	−0.349	C_15_H_22_N_4_O_3_	290sh, 320			*	1945	2	[56]
50	**MS2**: 771, 651, 609, 429, 357, 327 MS3: 357, 327, 299		Isoorientin 2”,6”-di-*O*-hexoside	[M−H]^−^	771.19946	771.1989	0.6895	C_33_H_40_O_21_	270, 344			*	75544	2	[61]
51	**MS2**: 489, **447**, 357, 327 **MS3**: 357, 327, 299	**MS2**: 593, **449**, 383, 329, 299 **MS3**: 431, 383, 353, 299	Isoorientin 7-*O*-glucoside	[M−H]^−^	609.1433	609.1450	−2.7680	C_27_H_30_O_16_	268, 348			*	75514	1	Std; [58]
52	**MS2**: 489, **447**, 327, 285, 255 **MS3**: 284, 226	**MS2**: **449**, 287, 269 **MS3**: **287**, 259, 213 **MS4**: 213, 153, 133	Luteolin 3′,7-di-*O*-glucoside	[M−H]^−^	609.1475	609.1461	2.285	C_27_H_30_O_16_	269, 343	*			75514	1	Std
53	**MS2**: 593 **MS3**: 285, 185, 153, 131	**MS2**: **595**, 491, 449, 335, 311, 287 MS3: 449, 287	Luteolin *O*-hexosyldeoxyhexoside-*O*-hexoside	[M−H]^−^	755.20408	755.204	0.084	C_33_H_40_O_20_	266, 349	*	*			3	[57]
54	**MS2**: **623**, 447, 315, 299 **MS3**: 357, 315, 299, 271, 255, 227	**MS2**: 657, 641, 625, 609, **479**, 317, 302, 273 **MS3**: 342, 317, 273	Isorhamnetin *O*-hexosyldeoxyhexoside-*O*-hexoside	[M−H]^−^	785.21381	785.2146	−0.982	C_34_H_42_O_21_	259, 369	*	*			3	[53,57]
55	**MS2**: 477, 357, **315**, 255, 217 MS3: 153		Isorhamnetin hexoside	[M−H]^−^	477.10435	477.1038	1.05	C_22_H_22_O_12_	252, 369	*	*			3	[53,57]
56	**MS2**: **315**, 255 MS3: 153		Isorhamnetin deoxyhexosylhexoside	[M−H]^−^	623.1624	623.1618	1.0327	C_28_H_32_O_16_	-		*		75752 or 75758	3	[57]
57	**MS2**: 575, **502, 473**, 413, 383 **MS3** (593-473): **383**, 353, 311 **MS3** (593-502): 413, 383, 312	**MS2**: **577**, 559, 541, **529**, 499, 457, 427 **MS3** (595-529): 511, 427, 367 **MS3** (595-577): 559, 529, 511, 481, 445, 427, 409, 380	Apigenin 6,8-di-*C*-hexoside	[M−H]^−^	593.1505	593.1512	−1.2503	C_27_H_30_O_15_	269, 339		*			3	[53,57]
58	**MS2**:561, 519, **489**, 459, 429, 399, 369 **MS3**:399, **369** **MS4**:341, 297	**MS2**: **563**, 545, 515, 497, 443, 413 **MS3**: **545**, 515, **497**, 485, 467, 395 **MS4** (563-497): 413, 395, 312 **MS4** (563-545): 509, 497, 467	Luteolin 6-*C*-pentoside-8-*C*-hexoside	[M−H]^−^	579.1492	579.1501	−1.489	C_26_H_28_O_15_	269, 348	*		*	69814	2	[62]
59	**MS2**: 399, **387**, 205, 181 MS3: 372, 203		Sinapoyl-homovanillic acid derivative	[M−H]^−^	597.18205	597.1825	−0.742		260, 335	*	*	*	75566	2	[58]
60	**MS2**: 609, 489, MS3: 489, 429, 309 MS4: 309		Isoorientin 2″-*O*-hexoside 7-*O*-[6″-sinapoyl]-hexoside	[M−H]^−^	977.26068	977.2568	3.929	C_44_H_50_O_25_	263, 340	*				3	[58]
61	**MS2**: 469, **307**, 161 **MS3**: 307, 161		Hydroxycoumarin hexoside-pentoside	[M−H]^−^	469.13626	469.1351	2.367	C_20_H_24_O_12_				*		3	[62]
62	**MS2**: **307**, 161, 145 **MS3**: 161, 145, 113		*p*-Coumaroyl-caffeic acid pentoside	[M−H]^−^	439.12554	439.1246	2.175	C_23_H_22_O_10_	286sh, 315			*		3	[56]
63	**MS2**: **163**, 135, 119 **MS3**: 119		*p*-Coumaroylthreonic acid	[M−H]^−^	281.0672	281.0667	1.864	C_13_H_14_O_7_	290sh	*				3	[56]
64	**MS2**: 561, 489, **459**, 399, 369, 327 **MS3**: 441, 399, **369** **MS4**: 341, 313	**MS2**: **563**, 545, 515, 497, 485, 473, 413 **MS3**: 545, **515, 473**, 449, 413, 365 **MS4** (563-473): 455, 437, 367, 341 **MS4** (563-515): 449, 431, 413	Luteolin 6-*C*-hexoside-8-*C*-pentoside	[M−H]^−^	579.1368	579.1355	2.17	C_26_H_28_O_15_	262, 345			*		3	[54]
65	**MS2**: 697, 535, 373, **329**, 178 **MS3**: 299, 284, 269, 178, 161		Hydroxypinoresinol di-*O*-hexoside	[M−H]^−^	697.23658	697.2366	2.377	C_32_H_42_O_17_	278	*		*	3421	2	[62]
66	**MS2**: 489, **429**, 327, 309, 285 **MS3**: **327**, 298 **MS4**: 297, 175		Isoorientin 6”-*O*-hexoside	[M−H]^−^	609.1440	609.1450	−1.6214	C_27_H_30_O_16_	266, 347	*			75353	3	[58]
67		**MS2**: 449, **303**, 285 MS3: 285	Quercetin *O*-deoxyhexosylhexoside	[M+H]^+^	611.1594	611.1607	−2.0245	C_27_H_30_O_16_	255, 353	*		*		2	[62]
68	**MS2**: 575, 533, **503**, 473, 431, 311 **MS3**: 413, 383, 311		Isovitexin 6”-*O*-hexoside	[M−H]^−^	593.15198	593.1520	1.3952	C_27_H_30_O_15_	268, 335	*				3	[53,57]
69	**MS2**: 533, 503, 473, **413**, 383, 341, 293 **MS3**: 312, **293**	**MS2**: **433**, 415, 397, 367, 337, 313, 283 **MS3: 415**, 397, **367**, 337, 313, 283367, 283 MS4:283	Isovitexin 2”-*O*-glucoside	[M−H]^−^	593.15094	593.15119	−0.42709	C_27_H_30_O_15_	268, 335	*				1	Std; [58]
70	**MS2**: 489, **429**, 369, 357, 339, 309 **MS3**: 429, 369, 351, 339, 309, 243	**MS2**: **449**, 431, 383, 353, 329, 299 **MS3**: 431, 413, **383**, 353, 329, 299 **MS4**: 299	Isoorientin 2”-*O*-glucoside	[M−H]^−^	609.1476	609.1461	2.45	C_27_H_30_O_16_	269, 348			*	17379	1	Std; [58]
71	**MS2**:429, 411, 357, **327**, 283 **MS3**: **297**, 283 **MS4**: 269	**MS2**: 431, **383**, 353, 329, 299 **MS3**: 299	Isoorientin	[M−H]^−^	447.0939	447.0933	1.4239	C_21_H_20_O_11_		*		*	17965	1	Std
72	**MS2**: 545, 503, 473, **443**, 413, 383, **353**, 325 **MS3** (563-353): 353, 325, 297 **MS3** (563-443): 383, 353, 297, 191	**MS2**: **547**, 529, 511, 451, 337 **MS3**: **530, 499**, 482, 458, 391 **MS4**: 512, 397	Apigenin 6-*C*-glucoside-8-*C*-arabinoside	[M−H]^−^	563.1409	563.1406	0.5600	C_26_H_28_O_14_	265, 335	*		*	17965	1	Std
73	**MS2**: 455, 503, 473, **443**, 383, 353, 337 **MS3**: **353**, 325, 297, 203 **MS4**: 325, 297		Apigenin 6-*C*-pentoside-8-*C*-hexoside	[M−H]^−^	563.1416	563.1406	1.7191	C_26_H_28_O_14_	266, 335	*	*	*	9047	2	[62]
74	**MS2**: 447, **357**, 327, 285 **MS3**: 339, 311, 297, 285	**MS2**: 431, **383**, 299 MS3: **299** MS4: 183, 121	Orientin	[M−H]^−^	447.0926	447.0933	−1.4845	C_21_H_20_O_11_		*		*	7781	3	[62]
75	**MS2**: 503, 473, 443, **383**, 353 **MS3**: 365, 325, 221	**MS2**: **547**, 500, 457 **MS3**: 511, 493, 409	Apigenin 6-*C*-pentoside-8-*C*-hexoside	[M−H]^−^	563.14198	563.1406	2.4	C_26_H_28_O_14_	266, 335				75589	3	[62]
76	**MS2**: 593, 503, 473, 431, **311**, 297, 283 **MS3**: 311, 283	**MS2**: **577**, 559, 529, 409, 475, 433, 415, 397, 367, 337, 313, 283 **MS3**: 559, **529**, 498, 415, 397, 367, 337, **283** **MS4**: 175	Isovitexin 7-*O*-glucoside	[M−H]^−^	593.1506	593.1215	−0.9416	C_27_H_30_O_15_	265, 335	*	*	*	75439	1	Std; [58]
77	**MS2**: 653, 491, 329 **MS3**: 315, 299		Tricin di-*O*-glucoside	[M−H]^−^	653.17377	653.1723	2.216	C_29_H_34_O_17_	266, 369		*			2	[58]
78		**MS2**: **463**, 301 **MS3**: 301, 286	Chrysoeriol di-*O*-hexoside	[M+H]^+^	625.1755	625.1763	−1.3123	C_28_H_32_O_16_	246, 266, 347		*			3	[57]
79	**MS2**: 574, 533, **503, 473**, 413, 383 **MS3**: **413, 383** **MS4**: 355, 312	**MS2**: **577**, 559, 529, 499, 463, 409, 356 **MS3**: 541, **529**, 499, 452, 427, 377 **MS4**: 427, 355	Chrysoeriol 6-*C*-hexoside-8-*C*-pentoside	[M−H]^−^	593.1518	593.1512	1.0147	C_27_H_30_O_15_	250, 348	*				3	[57]
80	**MS2**: 577, 503, **457**, 383, 353 **MS3**: 383, 353		Apigenin 6-*C*-hexoside-8-*C*-deoxyhexoside	[M−H]^−^	577.15643	577.1563	0.262	C_27_H_30_O_14_	266, 335	*		*		2	[62]
81	**MS2**: 574, 533, **503**, 473, 413, 383 **MS3**: **413**, 383 **MS4**: 352, 338, 312	**MS2**: **577**, 541, 457, 529, 511, 409, 389, 345 **MS3**: 559, **529**, 511, 427 **MS4**: 511	Chrysoeriol 6-*C*-pentoside-8-*C*-hexoside	[M−H]^−^	593.15277	593.1512	2.658	C_27_H_30_O_15_	246, 267, 346			*		3	[62]
82	MS2: **371**, 209, 175 MS3: 209, 121	**MS2**: 387, **373**, 369, 211, 193 **MS3**: 211	Blumenol *C*-hexoside-glucuronide	[M−H]^−^	547.23957	547.2396	−0.081	C_25_H_39_O_13_^−^	255	*	*	*		2	[58]
83	MS2: 341, **311**, 283 MS3: 283, 237, 117	**MS2**: 415, 397, **367**, 337, 283 **MS3**: 283, 271	Isovitexin	[M−H]^−^	431.0994	431.0984	2.4502	C_21_H_20_O_10_	268, 336			*	18330	1	Std; [58]
84	**MS2**: 503, **443**, 323 **MS3**: **323**, 308 **MS4**: 308	**MS2**: 607, 591, 542, **463**, 445, 397, 367, 343, 313, 265 **MS3**: 445, 427, 397, 367, 343, 313	Isoscoparin 2”-*O*-glucoside	[M−H]^−^	623.1611	623.1618	−1.1011	C_28_H_32_O_16_	250, 348	*		*	75518	1	Std; [58]
85	**MS2**: 491, 373, **329** MS3: 315, 175		Tricin hexosylmalonate	[M−H]^−^	577.12073	577.1199	1.45	C_26_H_26_O_15_	-	*		*	75518	3	[53]
86	**MS2**: 515, 473, 443, 413, 383, **353** **MS3**: 325, **297** **MS4**: 267	**MS2**: 517, 499, 481, 469, 433, 415, 397, 308 **MS3**:481, 463, 445, 433, 409, 379 **MS4**: 463, 445, 433, 397, 373, 351, 329	Apigenin 6-*C*-pentoside-8-*C*-pentoside	[M−H]^−^	535.1458	535.1446	2.2345	C_25_H_26_O_13_	265, 335	*	*			3	[53]
87	**MS2**: 371, **341**, 298 **MS3**: 327, 313, 298	**MS2**: **455**, 427, 409, **397**, 367, 343, 313 **MS3** (463-397): 379, 313, 301, 298 **MS3** (463-445): 427, 397, **367**, 313, 253 **MS4** (445-367): **339**, 324 **MS5** (367-339): 324, 311	Isoscoparin	[M−H]^−^	461.1089	461.1089	0.0285	C_22_H_22_O_11_		*		*	18200	2	[62]
88	**MS2**: 476, 329, 314 **MS3**: 314, 299	MS2: **331**, **MS3**: 315, 287, 270	Tricin 7-*O*-glucoside	[M−H]^−^	491.1192	491.1195	−0.6682	C_23_H_24_O_12_	266, 368	*		*	75349	1	Std
89	**MS2**: **329**, 314, 299 **MS3**: 314, 299	**MS2**: 493, **331**, 315 **MS3**: 315, 269	Tricin *O*-hexosyldeoxyhexoside	[M−H]^−^	637.1769	637.1774	−0.733	C_29_H_34_O_16_	265, 367	*	*	*	131777	3	[58]
90	**MS2**: **329**, 313 **MS3**: 314	**MS2**: **493**, 475, 331 **MS3**: **331**, 315 **MS4**: 315, 269	Tricin *O*-deoxyhexoside-*O*-hexoside	[M−H]^−^	637.19042	639.19196	−2.3961	C_29_H_34_O_16_	266, 367			*		3	[53,57]
91	**MS2**: 607, 299, 284	**MS2**: 463, 301	Chrysoeriol *O*-hexosyldeoxyhexoside	[M−H]^−^	607.16864 [M−H]^−^	607.1668	2.959	C_28_H_32_O_15_	249, 250, 345	*	*	*		3	[53,57]
92	**MS2**: **313**, 299 **MS3**: **299**, 285, 161 **MS4**: 271, 203, 161	**MS2**: **315**, 270, 253 **MS3**: 299, 270, 242, 207, 153	Tricin	[M−H]^−^	329.06758	329.0667	2.747	C_17_H_14_O_7_		*		*	59979	3	[53,57]
93	**MS2**: 383, 267, 249, 193, 134, 113		Feruloylhydroxycitric acid	[M−H]^−^	383.06252	383.062	1.4026	C_16_H_16_O_11_		*	*		176361	3	[58]

## Data Availability

Most of the data used in the current study is contained within the article. Remaining data that support the findings of this study is available from the corresponding authors upon request.

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
