# Peer review of "Targeted and Untargeted Metabolomic Analyses Reveal Organ Specificity of Specialized Metabolites in the Model Grass Brachypodium distachyon"

_molecules, 2022, doi:10.3390/molecules27185956_

Round 1

Reviewer 1 Report

The manuscript “Targeted and untargeted metabolomic analyses reveal organ specificity of specialized metabolites in the model grass Brachypodium distachyon” identify that differences in root metabolite profiles in Bd21 and Bd3-1 lines which differ strongly in their root morphology. The results revealed metabolic pathways that significantly differentiate analyzed organs and accessions. Based on the MS/MS and MSn analyses, 93 specialized metabolites mainly phenylpropanoids were identified in the model grass plant. The results are well presented with a lot of interesting information. However, there are some aspects of the manuscript that need to be modified and improved. The comments for the manuscript are as follows:

1. Latin names in the text should be in italic.

2. The character of Brachypodium Bd21 and Bd3-1 lines should be provided in Introduction;

3. Please provide Figure 1,2,3 figures with high-quality. Please provide the figures of Brachypodium Bd21 and Bd3-1 samples with the trait of leaves, root and spike.

4. In figure 3, please describe more concise figure legend of value and color.

5. In table 3, 93 metabolites were identified in leaves, roots and spikes of Brachypodium. The list is too long and if there is other form to exhibit the character of metabolites.

6. Line 359: H2O, line 361: CO2. Please check the style of other words in the text.

7. The references should be revised with the instruction of Molecules journal .

Author Response

Thank you very much for your appreciation of our work and for spotting some issues with our manuscript.

  1. As discussed with the editorial office there was a problem with converting our manuscript upon it was uploaded. Consequently some of the formatting (italics, superscript, subscript etc.) has been lost. We are sorry for this confusion. We have restored the formatting including italics for Latin names.
  2. We have expanded the information about Bd3-1 and Bd21 lines in the introduction. This includes additional references.
  3. High quality Figures 1, 2 & 3 have been uploaded independently from the figures embedded in the manuscript text. We hope they are of sufficient quality. Unfortunately, we do not have publishing quality photographs of collected Bd21 and Bd3-1 leaf, root and spike samples. To obtain such photographs we should grow Brachypodium plants till the stage we collected samples and this would take at least 10-12 weeks. Regarding the fact that we do not refer to any morphological differences between both lines we do not see it really necessary to include such photographs.
  4. Color scale presents log2 from peak heights of respective ions. We have indicated this in the figure and in the figure legend.
  5. The whole content of Table 3 is important for identification of listed compounds. This is especially important as Reviewer #2 questions some of the identifications. For this reason we would refrain from shortening Table 3. In case the editor considers Table 3 as too bulky we can move it to the Supplementary materials.
  6. This is the same issue as mentioned in reply to comment #1. We have restored now subscripts in chemical formulas.
  7. This discrepancy has been partially caused by inappropriate manuscript conversion. The reference list has been revised now according with the instruction of Molecules journal.

Reviewer 2 Report

The manuscript describes the metabolomic analysis of Brachypodium dystachyon. It seems quite complete in the different aspects discussed. Still, the approach for studying the flavonoid glucosides needs more support because the comparison with related species is not enough to assure the identity of the compounds.

Author Response

Thank you very much for your appreciation of our work. We do not base our compound identification on comparison with different species. We refer to other species to discuss if the same/similar compounds are present in related grasses. Compound identification is primarily based on fragmentation of protonated or deprotonated ions and postulated structures are supported with chemical formulas calculated from exact masses measured for protonated or deprotonated molecules as shown in Table 3. We inspected MSn and MS/MS spectra to guarantee in-depth and solid characteristics of specific glycosylation types of isomeric flavonoids, as well as structures of isomeric hydroxycinnamic acid conjugates and indolic compound conjugates. The sites of glycosylation and acylation on phenolic rings were characterized by ion trap mass spectrometry and observation of the multiple fragmentations in MSn analysis.

We fully agree with the Reviewer and make it clear in the manuscript that identity of many of the compounds is not fully confirmed and the proposed structures are preliminary. However, although preliminary, our identification strictly follows widely accepted standards of the Metabolomics Standards Initiative and we provide respective identification level (Sumner et al., 2007) for each structure in Table 3. Already at the beginning of our study we decided to preliminary identify a higher number of compounds to draft Brachypodium organular metabolom rather than to focus on validation of all structures, which would require NMR analysis (it is not realistic that we purify all compounds for which we do not have authentic standards and confirm their structures with NMR analysis). We strongly believe that even such preliminary identification is of significance for future functional genomic, phenotypic and physiological studies in Brachypodium.